

# Uncovering unseen fungal diversity from plant DNA banks

Erin M. Datlof[1], Anthony S. Amend[1], Kamala Earl[2], Jeremy Hayward[1,†], Clifford W. Morden[1], Rachael Wade[1], Geoffrey Zahn[1] and Nicole A. Hynson[1]

[1] Department of Botany, University of Hawai'i at Mānoa, Honolulu, HI, United States of America
[2] School of Forest Resources and Conservation, University of Florida, Gainesville, FL, United States of America
[†] Deceased.

## ABSTRACT

Throughout the world DNA banks are used as storage repositories for genetic diversity of organisms ranging from plants to insects to mammals. Designed to preserve the genetic information for organisms of interest, these banks also indirectly preserve organisms' associated microbiomes, including fungi associated with plant tissues. Studies of fungal biodiversity lag far behind those of macroorganisms, such as plants, and estimates of global fungal richness are still widely debated. Utilizing previously collected specimens to study patterns of fungal diversity could significantly increase our understanding of overall patterns of biodiversity from snapshots in time. Here, we investigated the fungi inhabiting the phylloplane among species of the endemic Hawaiian plant genus, *Clermontia* (Campanulaceae). Utilizing next generation DNA amplicon sequencing, we uncovered approximately 1,780 fungal operational taxonomic units from just 20 DNA bank samples collected throughout the main Hawaiian Islands. Using these historical samples, we tested the macroecological pattern of decreasing community similarity with decreasing geographic proximity. We found a significant distance decay pattern among *Clermontia* associated fungal communities. This study provides the first insights into elucidating patterns of microbial diversity through the use of DNA bank repository samples.

## INTRODUCTION

Understanding biodiversity is an important goal of biology. This is particularly critical in a changing world with habitat degradation and fragmentation, population declines, and species extinctions (*Vitousek et al., 1997*). Once a species becomes extinct, the genetic history resulting from its evolution is lost as well (*Mattick, Ablett & Edmonson, 1992*). DNA banks were initially developed to collect genetic material to create a storage base for evolutionary history, biological diversity, and genomic information (*Mattick, Ablett & Edmonson, 1992*). Throughout the world, samples are collected and stored in these banks to document and preserve genetic diversity (*Spooner & Ruess, 2014*). For extinct species, DNA bank samples act as storage deposits for their genomes (*Adams, 1994*; *Spooner & Ruess, 2014*).

Corresponding author
Nicole A. Hynson,
nhynson@hawaii.edu

In addition to the importance of DNA bank repositories for archiving target organisms' genetic information, these samples also harbor the microbial diversity associated with each accession. These samples represent well-preserved DNA at snapshots in time and space. For example, plant DNA bank samples not only preserve the genomic information of the target species, but also preserve potentially important cryptic microbial symbionts associated with the host, such as fungi known to inhabit the plant phyllosphere (*Porras-Alfaro & Bayman, 2011*; *Vorholt, 2012*).

Despite much work on patterns of plant diversity, comparatively little is known about the diversity of fungi. Fungi play crucial functions in ecosystems by acting as decomposers and nutrient cyclers, important mutualists such as mycorrhizae, and pathogens influencing host species populations (*Kendrick, 2001*; *Lips et al., 2006*). Globally, <100,000 species of fungi have been described (*Blackwell, 2011*), which is far less than total estimated fungal diversity, and also less than vascular plants, with <400,000 species currently described (*Royal Botanic Gardens Kew, 2016*). Estimates of global fungal species richness have increased almost 3–5 fold in the past 20 years, from 1.5 million (*Hawksworth, 1991*) to 3.5–6 million species (*O'Brien et al., 2005*; *Taylor et al., 2014*). This increase is due in part to advances in direct environmental DNA sequencing and extrapolations based on predictions of vascular plant to fungal ratios (*O'Brien et al., 2005*; *Taylor et al., 2014*). In order to obtain more accurate estimates of true fungal diversity, increased sampling using high throughput DNA sequencing of many different types of environments is needed, and DNA banks may significantly contribute to filling this knowledge gap.

Hawai'i is a biodiversity hotspot, making it an exceptional location to study patterns of species diversity (*Myers et al., 2000*). However, we know very little about Hawaiian fungi, their potential rates of endemism, and patterns of biodiversity. A survey of mushrooms throughout the Hawaiian Islands conducted in the 1990's found 310 species. The majority of these taxa were introduced, however 52 were putatively native and 46 of these taxa were considered potentially endemic (~88%; *Hemmes & Desjardin, 2002*). Similar rates of endemism are found in the Hawaiian flora.

An estimated 89% of the Hawaiian vascular plant flora is endemic (*Wagner, Herbst & Sohmer, 1999*). The unique Hawaiian flora is threatened by habitat degradation and loss, coupled with species invasions, which have led to native species becoming endangered or extinct (*Morden, Caraway & Motley, 1996*). There are currently 1,175 recognized native (endemic plus indigenous) Angiosperm species in Hawai'i (*Smithsonian Institution, 2017*) and 422 of these plants are currently endangered (35.9%; *US Fish and Wildlife Service, 2015*) with 104 taxa extinct or possibly extinct (8.8%; *Sakai, Wagner & Mehrhoff, 2002*). As a result of these extinctions and a strong potential for additional future losses, the Hawaiian Plant DNA Library (HPDL) was created to preserve the genetic diversity of the Hawaiian flora (*Morden, Caraway & Motley, 1996*).

The HPDL was started in 1992 and currently has over 10,000 accessions representing over 86% of all of the Hawaiian plant genera and over 50% of all species (*Morden, 2017*). Similar to other DNA banks across the world, collections for common species contain multiple individuals with their own accession number. However, the HPDL is unique due to the relatively disproportionate number of threatened and endangered native Hawaiian

plants compared to other localities, and thus serves as a genetic repository for many species that are endangered or extinct in the wild (*Morden, Caraway & Motley, 1996*). The main goal of the HPDL is to preserve Hawaiian plant DNA and bank samples for use in future studies of biodiversity (*Morden, Caraway & Motley, 1996*; *Randell & Morden, 1999*).

All naturally occurring plant tissues harbor fungi as both endophytes, living in between plant cells (*Rodriguez et al., 2009*) and epiphytes living on plant surfaces (*Santamaría & Bayman, 2005*), collectively known as phyllosphere fungi (*Vacher et al., 2016*). These communities form diverse assemblages with some studies showing an average of about 100 species per individual tree and ranges of about 700–4,000 species of fungi per host species (*Jumpponen & Jones, 2009*; *Zimmerman & Vitousek, 2012*). Thus, along with the HPDL's banked plant samples it also has likely and coincidentally preserved a substantial portion of the diversity of Hawaiian fungi, acting as a repository for not only plant genetics, but their microbiomes as well.

Utilizing several plant bank samples from the HPDL we investigated the diversity of unintentionally co-sampled fungi found within banked plant samples. Phyllosphere fungal communities can be cryptic and hyperdiverse (*Jumpponen & Jones, 2009*; *Zimmerman & Vitousek, 2012*) and this poses a challenge to studies with the goal of assessing microbial diversity. Even with the adoption of next generation DNA sequencing technologies, such as high throughput amplicon sequencing from environmental samples, observed diversity is often an under representation of true diversity (*Chiu & Chao, 2016*). Thus, rather than superficially sequencing the fungi from all available replicate DNA extracts from a single plant species, we chose to deeply sequence ten samples of a common endemic Hawaiian plant genus, *Clermontia* (Campanulaceae), with species found across the Hawaiian Islands (*Givnish et al., 2009*). This deep sequencing was done in hopes that the vast majority of fungi from our samples would be recovered from each sample.

In this study, we utilized historical DNA bank samples to validate the use of plant bank samples as a resource for elucidating phyllosphere fungal biodiversity, while subsequently examining plant-associated fungal diversity across space. Our two main questions were do DNA bank samples store microbial diversity? And can these previously collected samples be used to uncover ecological patterns, such as changes in microbial community similarity over space? Using DNA samples of eight species of *Clermontia* stored in the HPDL, we sequenced the fungi found in these plants' phylloplanes (the portion of the phyllosphere made up by leaves). We took advantage of the archipelago's geographic spatial gradient and the previously collected samples in the HPDL to test for decreases in community similarity as the distance between communities increases, the classical ecological pattern of distance-decay of community similarity (*Nekola & White, 1999*).

## MATERIALS & METHODS
### Samples
We examined foliar DNA extracts stored in the HPDL from the genus *Clermontia*, representing eight species collected from Hawai'i Island, O'ahu, Maui, Moloka'i and

**Table 1   DNA bank samples and accession numbers from the HPDL for each extract along with associated metadata.** In this study, two individual extracts were pooled for each location and given a sample code labeled by island.

| Sample | Sample code | HPDL number | *Clermontia* species | Island | Date extracted | Latitude | Longitude |
|--------|-------------|-------------|----------------------|--------|----------------|----------|-----------|
| 1<br>1a | M1 | 6,843<br>6,844 | *kakeana* | Moloka'i | 7/14/11 | 21.13 | −156.92 |
| 2<br>2a | H1 | 6,961<br>6,962 | *calophylla* | Hawai'i | 11/18/11 | 20.09 | −155.74 |
| 3<br>3a | H2 | 6,888<br>6,889 | *kohalae* | Hawai'i | 9/17/11 | 20.08 | −155.74 |
| 4<br>4a | H3 | 6,856<br>6,857 | *clermoniotides* | Hawai'i | 8/3/11 | 19.21 | −155.60 |
| 5<br>5a | H4 | 7,339<br>7,940 | *peleana ssp. singulariflora* | Hawai'i | 6/7/13 | 20.18 | −155.80 |
| 6<br>6a | K1 | 5,089<br>5,090 | *fauriei* | Kaua'i | 9/6/05 | 22.09 | −159.59 |
| 7<br>7a | O1 | 6,809<br>6,810 | *kakeana* | O'ahu | 6/17/11 | 21.34 | −157.82 |
| 8<br>8a | O2 | 7,008<br>7,009 | *oblongifolia ssp. oblongifolia* | O'ahu | 3/15/12 | 21.41 | −158.10 |
| 9<br>9a | Ma1 | 6,875<br>6,876 | *arborescens* | Maui | 7/5/11 | 20.82 | −156.28 |
| 10<br>10a | Ma2 | 6,831<br>6,832 | *kakeana* | Maui | 7/14/11 | 20.80 | −156.23 |

Kaua'i (Table 1; see Table S1 for more details). When initial collections were made, young leaf samples were collected in the field, sealed in bags, and stored at 4 °C until DNA extraction (less than a week). Leaves were not disturbed by rinsing prior to DNA extraction. Approximately 1.0 g of leaf tissue was extracted using a modified CTAB method with cesium chloride banding optimized to recover high quality DNA intended for long-term storage, and stored at −20 °C (*Doyle & Doyle, 1987*; *Morden, Caraway & Motley, 1996*). Samples used in this study were in a −20 °C freezer from 2–10 years. Two individual plant DNA extracts of each species per location were equally pooled, yielding a total of ten samples from 20 *Clermontia* individuals ($n = 10$), with *C. kakeana* replicates on three different islands: O'ahu, Moloka'i, and Maui.

## PCR and sequencing

These pooled extracts were individually prepared for fungal DNA sequencing with slight modifications to the Illumina 16S Metagenomic Sequencing Library Preparation protocol using a two-step PCR and index attachment (*Illumina, 2013*). Fungal DNA amplicons of the ∼250–400-bp targeted nuclear ribosomal Internal Transcribed Spacer 1 (ITS1) locus were amplified using ITS1F primers with Illumina adapter overhangs (5′ Adapter-CTTGGTCATTTAGAGGAAGTAA-3′; *Gardes & Bruns, 1993*) and modified ITS2 primers (5′ Adapter-GCTGCGTTCTTCATCGATGC-3′; *White et al., 1990*). The ITS locus is the official fungal DNA barcode (*Schoch et al., 2012*). Amplicons were purified and size-selected

using SPRIselect beads (Beckman Coulter, Inc., Brea, CA, USA), followed by a second PCR attaching forward and reverse eight-base pair barcoded Illumina overhang adapters (i7 and i5; *Illumina, 2013*). See Table S2 for PCR recipes and thermalcycler parameters. These indexed libraries were bead purified and quantified using the Qubit dsDNA HS kit (Life Technologies Inc. Gaithersburg, MD, USA). Libraries were then pooled at equimolar concentrations and sent to the Hawai'i Institute for Marine Biology Genetics Core Facility (HIMB) for quality control on the Agilent 2100 Bioanalyzer (Agilent Technologies, Santa Clara, CA, USA) and sequencing on the Illumina MiSeq platform v.3 paired-end 2 × 300 (Illumina, San Diego, CA, USA).

## Bioinformatics

De-multiplexed fastq files were obtained from the sequencing facility from the ten *Clermontia* plant bank samples. Raw sequencing data was deposited to the National Center for Biotechnology Information Sequence Read Archive (NCBI SRA) under BioProject PRJNA379349. These paired-end reads were merged with the Illumina Paired-End reAd mergeR (PEAR), keeping reads with a minimum assembly length of 250-bp, average quality threshold of 15 and above, and discarding all reads with any uncalled bases (*Zhang et al., 2014*). Further quality control was carried out using the FASTX-Toolkit, using the fastq_quality_filter command (*Hannon Lab, 2016*), where all reads with any base pairs containing a quality score below 15 were discarded (*Hannon Lab, 2016*). Potential chimeras were removed in vsearch (*Rognes et al., 2016*) using the uchime_ref command (*Edgar et al., 2011*), which referenced the User-friendly Nordic ITS Ectomycorrhiza (UNITE) database, accessed on 11.03.2015 (*Kõljalg et al., 2013*). Operational Taxonomic Units (OTUs) were clustered within QIIME (*Caporaso et al., 2010*) using the open-reference method (*Navas-Molina et al., 2013*) following the Usearch algorithm (*Edgar, 2010*). Briefly, reads were matched to reference OTUs in the UNITE dynamic database (ver7) (*Kõljalg et al., 2013*) with added *Clermontia* outgroups, then remaining reads that failed to match were subsampled as seeds for three subsequent rounds of *de novo OTU-picking*. The most abundant sequence for each OTU was chosen as a representative sequence. Singleton reads were removed in QIIME prior to OTU table generation and taxonomy was assigned against the UNITE database with the Basic Local Alignment (BLAST) algorithm.

## Statistics

All statistical analyses were conducted in R version 3.3.0 (*R Core Team, 2017*). The OTU table from QIIME was imported into R with the package *biomformat* (*McMurdie & Paulson, 2016*). OTUs that mapped to plant taxonomies or those that had no BLAST hit were removed from the OTU matrix and all OTUs with greater than ten reads were kept for analyses. Samples were rarefied to 16,546 reads, the minimum sample depth. Rarefaction, species accumulation curves were generated using the *vegan* package for: all samples, individual samples, and samples pooled by island (*Oksanen et al., 2017*). Because observed species richness often under estimates true species richness (*Hughes et al., 2001*), asymptotic extrapolations of species richness and diversity for all samples and species were estimated based on the first three Hill numbers using the *iNEXT* package for raw incidence

data (*Hsieh, Ma & Chao, 2016*). These are namely species richness, the exponential of Shannon entropy, and the inverse Simpson concentration, represented by $q = 0, 1, 2$, respectively (*Chao et al., 2014*). Hill numbers and extrapolations were generated based on individual samples and individual species. A Venn diagram was generated to visualize overlapping taxa between islands using the *VennDiagram* package (*Chen, 2016*).

## Distance matrices

To investigate ecological patterns, we accounted for variables that may be influencing the fungal communities found in these banked samples. These factors were temporal and physical distances between sample collections, as well as fungal community dissimilarity. Pairwise distance matrices were calculated for physical distance in kilometers using the *geosphere* package (*Hijmans, 2016*), time between sample collections in days, and Bray-Curtis community dissimilarity using the *vegan* package (*Oksanen et al., 2017*). Separate Mantel tests for each combination of the following pairwise distance matrices: time between sample collections (days), as well as Euclidean physical distance between samples (km), and community dissimilarity (Bray-Curtis), were run for 10,000 permutations. To investigate the effects of these variables a final partial Mantel test for physical distance and community dissimilarity, while controlling for time, was run for 10,000 permutations (*Oksanen et al., 2017*).

# RESULTS

## Sequencing

A total of 4,312,473 sequence reads were obtained from the plant DNA library samples. Of these, 3,571,252 paired-end reads (82.8%) were successfully assembled and further quality control removed low quality reads, keeping 2,680,945 reads (75.1%). After referencing UNITE, 121,618 (4.5%) chimeric sequences were removed, leaving a total of 2,559,327 high-quality reads. Taxonomic assignment yielded a total of 1,648,971 fungal reads that were binned into 2,944 fungal OTUs for use in in downstream analyses.

## Fungal diversity and host associations

Each *Clermontia* DNA bank sample used in this study contained fungal DNA. In total, we found 2,944 fungal OTUs associated with the ten foliar DNA bank samples. After removing OTUs with less than ten reads and rarefying to the sample with the minimum number of reads, we removed 1,164 OTUs (39.5%) and were left with a total of 1,780 fungal OTUs. While the observed OTU accumulation curve for all ten samples combined did not reach an asymptote (Fig. S1), OTU accumulation curves by sample and by island (except in the case of *C. fauriei* from Kaua'i and *C. kakeana* Moloka'i) generally reached their asymptotes, indicating that overall, we successfully recovered the majority of fungi present in our samples (Figs. S2 and S3). Observed richness per sample after rarefying, ranged from 108 to 682 fungal OTUs with an average of 295 OTUs per sample (±54.69 standard error). From our ten samples combined the *iNEXT* extrapolation curves suggest fungal richness based on the Hill number $q = 0$ (Chao1 richness) will saturate around 3,947 OTUs which would require at least 50 samples. Similarly fungal diversity based on $q = 1$ (exponential

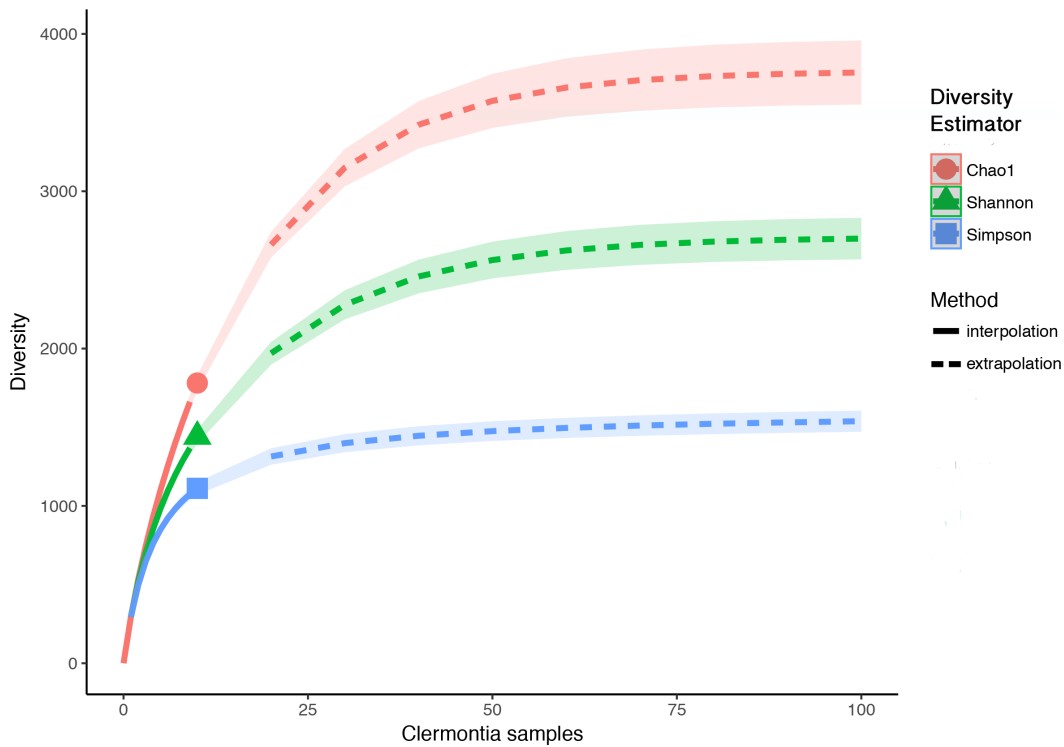

**Figure 1** **Sample interpolation (solid lines) and extrapolation (dashed lines) curves for phylloplane fungi from all ten *Clermontia* DNA bank samples.** Based Hill's numbers three different diversity estimators were used (Chao1 richness, exponential of Shannon entropy, and inverse Simpson concentration indices) and are shown by the different colors with 95% confidence intervals shown by shading. Shapes represent observed phylloplane fungal OTU diversity for the ten samples of *Clermontia* spp.

Shannon entropy) was estimated to saturate at around 2,750, and diversity based on $q = 2$ (inverse Simpson concentration) was estimated to saturate at about 1,591 (Fig. 1).

We investigated patterns of fungal diversity at the phyla and ordinal levels. Overall, the majority of fungi in the subkingdom Dikarya dominated all of the phylloplane samples, with phylum Ascomycota being most abundant followed by Basidiomycota (Fig. 2). Fungi belonging to the phylum Chytridiomycota and Zygomycota were also present in lower abundances. Additional OTUs mapped to kingdom Fungi but could not be identified further (Unidentified; Fig. 2). The top ten most abundant orders were Capnodiales, Chaetothyriales, Exobasidiales, Peltigerales, Pertusariales, Pleosporales, Tremellales, Ustilaginales, and two unknown orders (Fig. 3).

Based on our observed data, total average fungal OTU richness by island was 507.6 (±128.46 standard error). Oʻahu had the highest total observed richness with 1,045 OTUs, followed successively by Hawaiʻi (926 OTUs), Maui (685 OTUs), Molokaʻi (362 OTUs), and Kauaʻi again had the lowest richness (108 OTUs), though unlike the other islands we did not saturate our OTU richness by sequencing effort curve for *C. fauriei* from Kauaʻi and *C. kakeana* from Molokaʻi indicating that these are an underrepresentation of fungal OTU richness (Fig. S3). Overall about twenty OTUs were found on all of the five islands (Fig. 4).

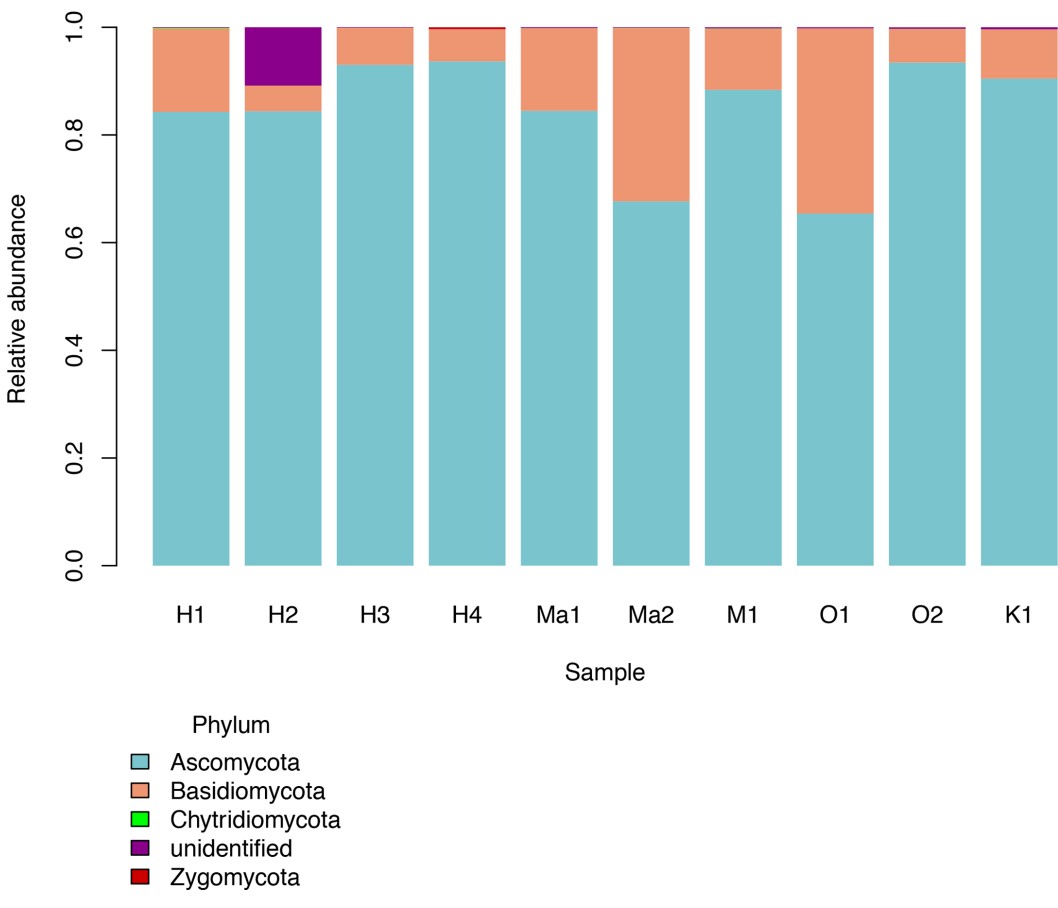

**Figure 2** Relative abundances of fungal phyla for each *Clermontia* DNA bank sample.

## Physical distance decay

*Clermontia* DNA bank extracts used in this study spanned across the main Hawaiian Islands. The nearest samples were collected less than one kilometer apart from a single site in Kohala, Hawai'i Island, and the furthest distance was 524.78 km from Kohala, Hawai'i Island to the Alaka'i Swamp, Kaua'i. Over this spatial range, while taking into account time (number of days) between sample collections, the fungal phylloplane communities exhibit a significant decrease in community similarity across increasing geographic distance (Fig. 5, Partial Mantel test: $r = 0.423$, $p = 0.005$). Time was also significantly correlated with physical distance between sample sites (Fig. S4, Mantel test: $r = 0.455$, $p = 0.048$). Time between sampling and community dissimilarity was marginally significantly correlated (Fig. S5, Mantel test: $r = 0.619$, $p = 0.051$). Despite the significant relationships with sampling time the Partial Mantel between distance and community dissimilarity while accounting for time, was significant (Fig. 5).

## DISCUSSION AND CONCLUSION

In this study we investigated the diversity of phylloplane fungi associated with *Clermontia spp.* that were collected across the Hawaiian Islands and stored as DNA bank samples.

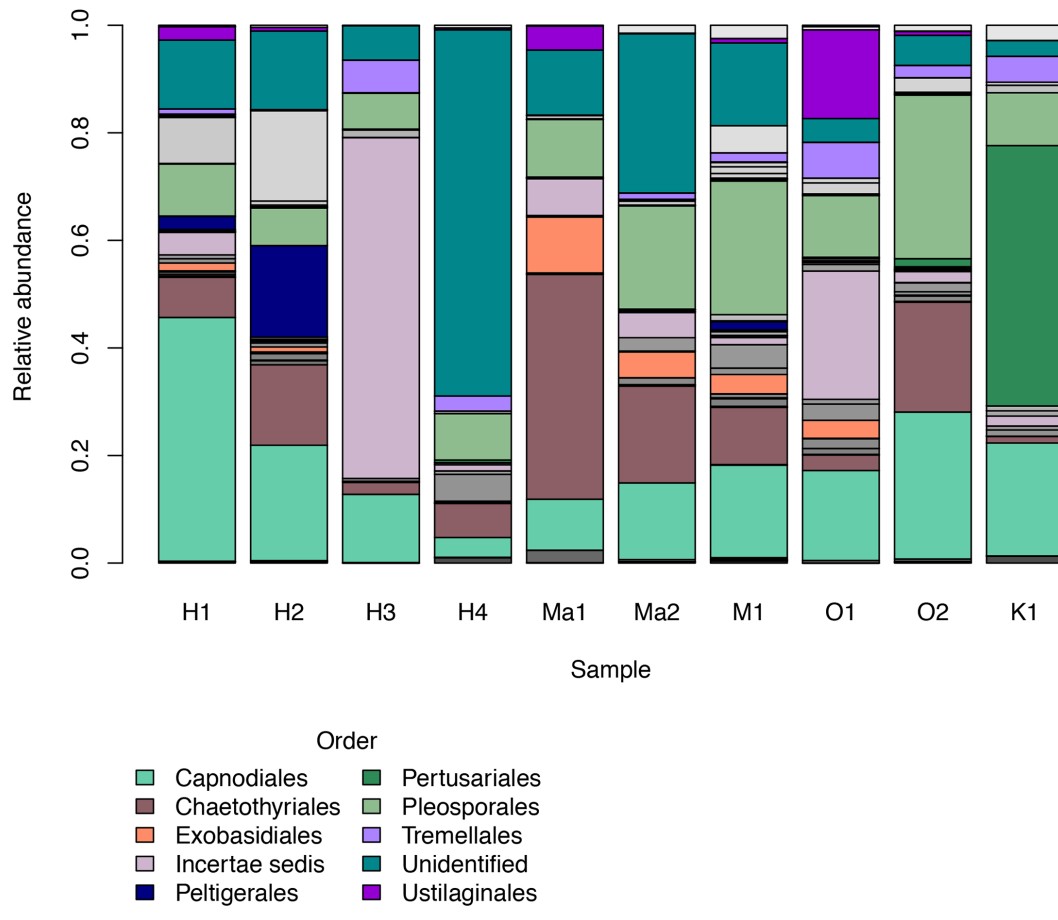

**Figure 3** **Relative abundances of the top ten most abundant fungal orders for each *Clermontia* DNA bank sample represented by color bars.** The less abundant orders are represented by grayscale bars.

While sample collection methods were not developed with the intention of preserving or analyzing phyllosphere fungi, they coincide with common practices for these purposes (*Jumpponen & Jones, 2009*; *Zimmerman & Vitousek, 2012*).

We found that these specimens harbored a considerable diversity of fungi. After quality control, we found 1,780 fungal OTUs from just ten samples, representing 20 *Clermontia* individuals and eight species. Fungal richness ranged from 108 to 682 OTUs per plant sample. Incredibly, this diversity was recovered from a total of just 20 grams of leaf tissue from which DNA was extracted and preserved. Despite our high sequencing depth, the observed species accumulation curves for all samples and islands did not saturate, indicating our sequencing efforts likely underestimated true *Clermontia* phylloplane fungal diversity. However, this novel use of DNA bank samples revealed substantial undiscovered fungal biodiversity stored in plant samples. These results provide further evidence of microbes making up the "unseen majority" of biodiversity (*Whitman, Coleman & Wiebe, 1998*), where a single macroorganism associates with a multitude of microorganisms both within and on their surfaces (*Turner, James & Poole, 2013*).

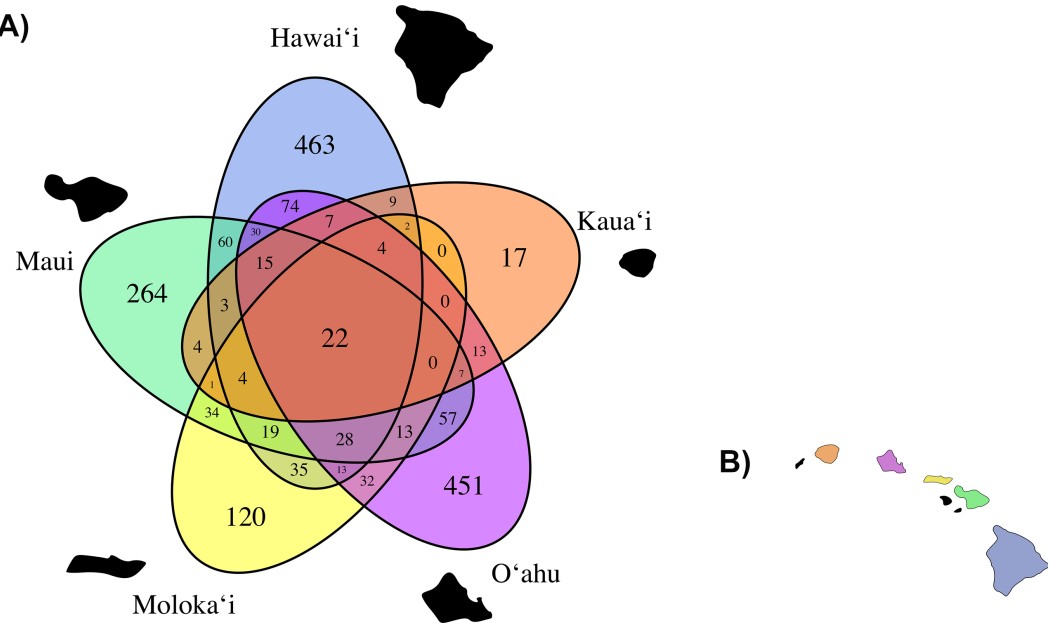

**Figure 4** (A) A Venn Diagram displaying the number of overlapping fungal OTUs shared between *Clermontia* samples from each of the five main Hawaiian Islands, represented by a different color. The number of OTUs unique to each island lie on the outermost portion of each ellipse. (B) The geographic layout of the main Hawaiian Islands.

This study highlights a new and underutilized function of biological collections, as well as gives insights into regional fungal diversity patterns. Previous estimates of total regional fungal richness have been based off of plant to fungi ratios ranging from 1:6 (*Hawksworth, 1991*) to 1:17 (*Taylor et al., 2014*). Our data supplement these studies using environmental NGS data. If we assume that the diversity of phylloplane fungi associated with *Clermontia* species is representative of the native Hawaiian flora, we would estimate based on Chao1 richness ($q = 0$) extrapolations (determined by species; Fig. S6) that the entire Hawaiian flora (c. 1,000 species) harbors about 4,000 fungi. This results in an approximate 1:4 plant to fungi species ratio. However, just considering phylloplane fungi associated with a single genus is likely an underestimate of total fungal biodiversity due to potential host-fungi specificity (*Hoffman & Arnold, 2008*). Supplementary to host specificity, only taking into consideration phylloplane fungi likely underestimates total regional fungal richness due to habitat niche partitioning among fungal species and guilds (*Hibbett, Gilbert & Donoghue, 2000*).

In addition to the study of microbial diversity, questions regarding microbial biogeography, host specificity, and the effects of global change on microbial communities could be addressed with DNA banks. For example, we were able to confirm the distance decay of microbial community similarity from DNA bank samples collected across the Hawaiian Islands. This finding is similar to other microbial systems where significant distance decay patterns were found in foliar endophytic (*Vaz et al., 2014*) and ectomyorrhizal fungal communities (*Bahram et al., 2013*), as well as bacteria and archaea (*Barreto et al., 2014*). We were also able to identify a temporal partitioning of phylloplane

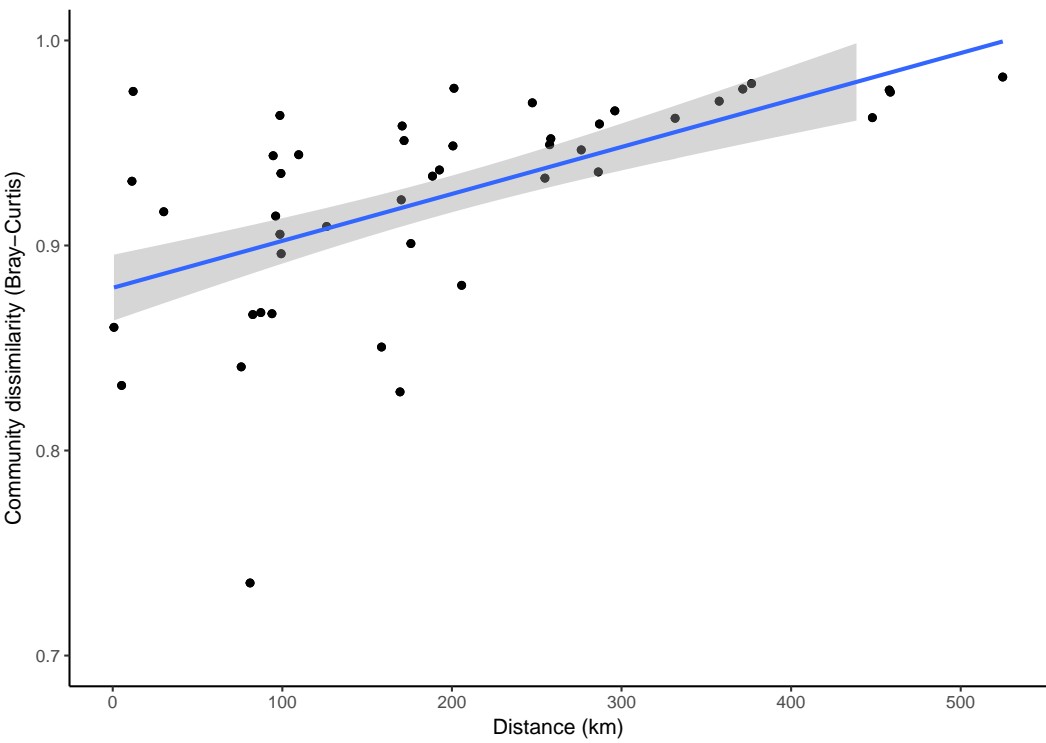

**Figure 5  Pair-wise Bray-Curtis fungal community dissimilarity plotted against corresponding pair-wise physical distances for each *Clermontia* plant bank sample spanning the main Hawaiian Islands.** A regression line was fit to the data, shown in blue, with 95% confidence intervals shown in grey. (Partial Mantel test: $r = 0.424$, $p = 0.005$, accounting for time between sample collection in days).

fungal communities (Fig. S5), indicating that DNA bank collections may be used to study changes in microbial communities over time. However, in addition to geography and temporal factors, taking into account host genotype, age, specificity, and differences in environment, such as light exposure and humidity, may potentially explain additional variation in fungal communities (*Hoffman & Arnold, 2008*).

In agreement with other phyllosphere studies, the majority of fungal taxa were identified as belonging to the subkingdom Dikarya, with the majority in phylum Ascomycota followed by Basidiomycota (*Rodriguez et al., 2009*). It is not surprising that we found so many unknown fungal taxa (45.16% of total OTUs at the family level) including 28 OTUs we were unable to place at the phylum level. The plant samples from this study represent an endemic Hawaiian genus whose microbial associates are previously unstudied, and possibly associate with undescribed fungi endemic to Hawai'i. However, this degree of unassigned fungal OTUs is not unique to our system and highlights our limited current knowledge of fungal diversity (*Nilsson et al., 2016*). For example, recent discoveries using environmental DNA sequencing have reshaped the fungal tree of life by uncovering a new fungal Phylum, the Cryptomycota (*Jones et al., 2011*). This stresses the need for further investigations of fungal biodiversity, their cryptic nature and diverse functions make for intriguing new discoveries that have the potential to change evolutionary and ecological theories based primarily on macroorganisms.

With the recent advent of next generation sequencing (NGS) techniques genomic investigations of non-model organisms have become readily accessible (*Da Fonseca et al., 2016*). However, there are important caveats to consider when using these methods and analyses. For example, working with environmental samples poses the challenging prospect of encountering hyperdiverse microbial communities such as the fungi found here, and other studies of plant phyllosphere fungi (*Arnold, 2007*; *Arnold & Lutzoni, 2007*). As seen in this NGS study, thousands of fungi can be associated with a small number of plant leaf samples. While uncovering this diversity is a goal of some microbial ecologists, for researchers using NGS techniques focused on the host organism (in this case plants), microbial symbionts may interfere with downstream analyses and results. Microbial taxa associated with macroorganismns should be taken into account when using NGS methods such as RAD seq, RNA seq, targeted sequencing, among other techniques (*Da Fonseca et al., 2016*). Additionally, future research into the microbiomes of hosts preserved in DNA banks should take into consideration how sample processing and storage may affect microbes. When initially collecting samples for these purposes, care should be taken to avoid microbial contamination from non-target hosts or environments.

Most DNA bank samples likely harbor unintended microbial communities associated with each target individual from a specific location at distinct snapshots in time. While DNA banks are a common genetic biodiversity repository (*Seberg et al., 2016*), to the best of our knowledge this is the first study where they were used to investigate genetic material other than that of the target organism. By using these archived samples we were able to rapidly recover previously undocumented microbial diversity. The abundance of DNA bank samples stored throughout the world represent a large proportion of the globes extant and extinct biological diversity. This storage provides the opportunity for microbes associated with these organisms to be easily investigated without the associated costs of sample collection. This may be important for conservation efforts, giving insight into potentially important symbionts (*Van der Heijden, Bardgett & Straalen, 2008*; *Busby, Ridout & Newcombe, 2016*). For those species that go extinct, their genomes are preserved in DNA banks along with their corresponding microbial symbionts. For extant organisms, DNA bank samples could be used to better understand the ecology of symbiosis and possibly identify coevolutionary patterns. Overall, this study highlights the potential use of DNA bank samples for the study of global biodiversity. This study also demonstrates the benefits of in-depth sample sequencing to uncover the majority of fungal diversity found in each plant bank sample. With DNA bank samples stored throughout the world, already collected, processed, and extracted, they harbor the potential for new and exciting investigations.

## ACKNOWLEDGEMENTS

We thank Emily Johnston and Richard O'Rorke for experimental and lab help, and Leah Tooman for further lab assistance. Also thanks to Sean Swift, Laura Tipton, and Sofia Gomes for R help. Thanks to Mitsuko Yorkston and Richard Pender for sample discussions. We would also like to thank Ricardo Araujo and two additional anonymous reviewers for their helpful suggestions on a previous version of this manuscript.

### Funding

This work was supported by a donation from the Illumina Corporation and National Science Foundation (No. 1255972) to Anthony S. Amend. The funders had no role in study design, data collection and analysis, decision to publish, or preparation of the manuscript.

### Grant Disclosures

The following grant information was disclosed by the authors:
Illumina Corporation and National Science Foundation: 1255972.

### Competing Interests

Anthony S. Amend is an Academic Editor for PeerJ.

### Author Contributions

- Erin M. Datlof conceived and designed the experiments, performed the experiments, analyzed the data, wrote the paper, prepared figures and/or tables, reviewed drafts of the paper.
- Anthony S. Amend conceived and designed the experiments, contributed reagents/materials/analysis tools, reviewed drafts of the paper.
- Kamala Earl and Rachael Wade conceived and designed the experiments, performed the experiments, reviewed drafts of the paper.
- Jeremy Hayward conceived and designed the experiments, analyzed the data, prepared figures and/or tables.
- Clifford W. Morden reviewed drafts of the paper and provided DNA bank samples.
- Geoffrey Zahn analyzed the data, prepared figures and/or tables, reviewed drafts of the paper.
- Nicole A. Hynson assisted with data analysis, wrote the paper, reviewed drafts of the paper.

### Data Availability

The raw sequence reads are deposited in NCBI Sequence Read Archive as BioProject PRJNA379349.

### Supplemental Information

Supplemental information for this article can be found online at http://dx.doi.org/10.7717/peerj.3730#supplemental-information.

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
