# Peer review of "Uncovering unseen fungal diversity from plant DNA banks"

_PeerJ, doi:10.7717/peerj.3730_

## Round 0.1 · original submission · Major Revisions

I have gone through the reviewers comments and recommendations which I consider exhaustive. I agree with all of them (many are concordant between at least two and some are unanimous) and in my opinion, although requiring a deep revision of the manuscript are likely to be solvable. Please note that a major concern is that aims and conclusions exceed the data support. A more modest and cautious formulation of both aims and conclusions is therefore mandatory.

Reviewer 1 ·

Basic reporting

Overall, the manuscript was written in a clear, professional and unambiguous language.
The authors should reconsider, throughout the manuscript, the use on the concept “phyllosphere”. This term refers to the above-ground part of the plant that harbors microorganisms. In the present work, DNA extract were obtained from leaves and, therefore, the concept “phylloplane” is more suitable.
The Introduction is well balanced and covers all the information needed for the interpretation of the results. Nevertheless, to improve clarity, please consider revising the text in lines 92-93.
The Material and Methods should be carefully revised according to the suggestions presented the Experimental design below.
In the Results, the number of decimal places must be consistent (line 224).
In the Discussion and Conclusion please confirm the number of fungal richness. In this chapter the authors wrote 686 while in the results, they wrote 682.

Experimental design

The major issue about the manuscript is the experimental design. The Hawaiian Archipelago is composed by thousands of islands. In the present work, only the main five islands were considered.
Another matter of concern the samples collection dates. Microorganism present a seasonal behavior; in the present work, samples were collected in different time points (ranging from March to November).
Additionally, the host plant differed amongst islands even the rationale behind the species and the number of plants chosen in each island is not clarified by the authors.
I am not sure if this sampling effort is enough to unravel the true fungal diversity of the Archipelago. The authors should consider choosing same species in all islands, calculate the adequate number of samples in each island, and collect samples during one full year (monthly collections).
In lines 205 and 206 the authors, state “there is a far more fungal diversity to uncover with increased sampling”, highlighting the reviewer’s concerns.

Validity of the findings

I do not understand the conclusion in Line 224 to 227. How is possible to make such a comparison? Different species and different number of samples were used in each island.

Additional comments

The manuscript " Not just contaminants: Uncovering unseen microbial biodiversity from plant DNA banks," by E M Datlof, A S Amend, K Earl, J Hayward, C W Morden, R Wade, G Zahn, N A Hynson, gives an innovative approach to the study of the microbial diversity of Hawaiian Island using samples collected from a germplasm bank.
The current works presents several aspects that should be improved upon before publication.

·

Basic reporting

The manuscript is clear and unambiguous. The structure was well-planned but there is repetition of figures on the manuscript and supplemental material. In addition, some figures are not necessary. The references are generally adequate but few additional details should be provided on DNA banks maintenance, specific objectives and biodiversity already known.

Experimental design

The primary research is within the scope of the journal. The research question was well-defined and it is relevant. There are some concerns on this research:
1) It should be clear to the readers what was the main proposes for building the DNA banks monitored on this study. How were the banks built? How are maintained? What diversity of plants was covered in each case? Are these DNA banks similar to other DNA banks worldwide and what specific differences can be found in these ones?
2) One critical cleaning step for OTU diversity is the identification of chimeras. Uchime2 is a new version of old uchime script and identifies more chimeras. The new version should be tested, or in case the authors keep the older version, a justification should be provided. The authors found 4.5% of chimera sequences that represents really low number (some authors suggest the true value may be around 40-50% depending on the sample).
3) What are the major confounders of this study? How can be corrected in the future?
4) Some figures are irrelevant (e.g. figure 3) and others are repeated in the supplemental material.
5) Do any of the fungal groups represent novelty in such samples? What is the fraction described in the natural environments?
6) Do the authors believe what is found in the DNA banks correspond entirely to what is found in nature? Is it possible that some deviations be found and why?
7) Preservation/storage of material in DNA banks can affect the microbial diversity measurements in the samples? The main propose of the DNA banks is to keep plant DNA but I agree that there room for some interesting studies considering microbial populations. Which are the major risks for microbial contamination of these banks? How is the quality of the material monitored?

Validity of the findings

Some concerns on the validity of the findings are described above. Some questions mainly concerning the quality of the material for microbial studies, risks for contamination and correspondence to what is found in the natural environment need to be clarified.

Additional comments

The study is interesting and may reveal some new insights. But questions mainly concerning the quality of the material for microbial studies, risks for contamination and correspondence to what is found in the natural environment need to be clarified.

Reviewer 3 ·

Basic reporting

The manuscript “Not just contaminants: Uncovering unseen microbial biodiversity from plant DNA banks” focuses on the utility of plant DNA banks to unveil diversity of phyllosphere associated fungi. Although the idea is meritorious the manuscript is not hypothesis-driven which impairs its novelty and holds major weaknesses that the authors should address:
- The title is misleading since only fungal diversity has been assessed, and not the whole phyllosphere microbiota, which includes bacteria most certainly highly represented. Furthermore by writing: “Not just contaminants” the authors implicitly assume that the Clermontia associated-microbiota are unwanted constituents of the sample and not part of total DNA samples from Clermontia leaves.
- The manuscript discusses the phyllosphere fungal diversity of the endemic Hawaiian plant genus Clermontia having into account the geographic distance between samples, but neglecting the plant species variable, regardless the fact that samples of 8 Clermontia species have been analyzed. In fact, it is not clear why the study did not focus on the phyllosphere microbiota of a single Clermontia species across different islands.

Experimental design

This study provides a characterization of phyllosphere fungal diversity inferred by a NGS approach in 10 samples from 8 Clermontia species across 5 different Hawaiian islands. Three samples were obtained from a single species (C. kakeana) in three different Islands; and only a single sample was analyzed for two islands. Although these studies are important, the sampling strategy did not seem to have been planned according a research driven hypothesis, therefore hindering robust interpretation of the results and compromising the novelty of this study.

Validity of the findings

At some instances the interpretation of the results is unsatisfactory and the conclusions are not always sustained by the data:
i) As the authors stated the number of Clermontia samples stands below the minimal needed to determine assertively fungal diversity as inferred by rarefaction curve analysis (pp.9, ln 204-208). Probably the authors would manage to get robust rarefaction curves if they have focused on 10 samples from a single Clermontia species, from 2 to 3 different islands.
ii) There is some redundancy in the Tables and Figures. For instance Table 1 and Table S1, repeat most of the data; Fig. 4 provides basically the information already available on Fig. S1; Fig. 1 and Fig 2 provide the same data at different taxonomic levels for the different samples, which is further detailed in Fig. 3, where OTUs from all the samples are sum up. Fig. 3 as it is does not bring any added-value. The information provided in Table S2 could be easily integrated as a text within M&M.
iii) The authors need to revise the graphs regarding the units: it is not understandable why in Figs S1, S3 and S4 have as Y-axis unity “OTUs” and Fig. 4 “Diversity”.
iv) The authors determine that fungal diversity ranged from 108 to 686 OUT per plant sample (pp. 11, ln243). These numbers are inconsequential since they are certainly biased by the insufficient number of reads as inferred by Fig. S1, S2 and S3;
v) The authors should have explain why “time was not a significant predictor of community” (pp. 12, ln270-272), regardless the fact that samples were not collected in the same year or season;
vi) Although it is expectable to find unknown fungal taxa, it is a bit intriguing why sample H2 is characterized by a considerable number of unidentified fungi at the phylum level, in comparison to other samples (Fig. 1). The authors should have hypothesized about this.

---

## Round 0.2 · Minor Revisions

I consider that most of the significant points raised by reviewers on the first version were successfully addressed.

From the issues raised by the reviewers on the resubmission, I retain just a few that I also consider requiring your attention before being able to accept your manuscript for publication:

Rev#1
Line 42: Understanding biodiversity is an important goal of biology and ecology.
I suggest (since Ecology is a branch of Biology), to keep just: Biology
Line 66: These increases are due
Please contemplate changing the text to: This increase is due
Lines 120 – 132: the text written by the authors should not be included in the Material and Methods chapter. Please considering removing or moving it to other chapters (such as Introduction and/or Discussion).
Lines 140 – 143, 198 – 201: the text written by the authors should not be included in the Material and Methods chapter. Please considering removing or moving it to other chapters (such as Introduction and/or Discussion).
Line 569: please include the HPDL acronym in the legend of Table 1.
Rev#2
I would like the objections 1 to 4 raised under Experimental design to be answered.
Typos:
Mantel instead of mantel

Reviewer 1 ·

Basic reporting

1. Basic Reporting:

As highlighted in the first revision of the present work, overall, the manuscript was written in a clear, professional and unambiguous language. Although some minor aspects should be addressed to improve manuscript’s clarity:

Lines 25 - 26: DNA banks are used as storage repositories for genetic diversity of organisms ranging from plants to insects to mammals throughout the world.
Please contemplate changing the text to: Throughout the world, DNA banks are used as storage repositories for genetic diversity of organisms ranging from plants to insects to mammals.

Lines 27 – 28: These banks preserve the genetic information for organisms of interest, however they also indirectly preserve organisms’ associated microbiomes, including fungi associated with plant tissues.
Please contemplate changing the text to: Designed to preserve the genetic information for organisms of interest, these banks also indirectly preserve organisms’ associated microbiomes, including fungi living and colonizing plant tissues.

Lines 37: please contemplate replacing historic samples by historical samples.

Line 42: Understanding biodiversity is an important goal of biology and ecology.
Please contemplate changing the text to: Understanding biodiversity is an important goal of both Biology and Ecology.

Line 48: initially developed to collect genetic material in order to create a storage base for evolutionary
Please contemplate changing the text to: initially developed to collect genetic material to create a storage base for evolutionary.

Line 55 - 59: These samples represent well-preserved DNA at snapshots in time and from specific locations. For example, plant DNA bank samples not only preserve the targeted species’ genomic information, but also preserve potentially important cryptic microbial symbionts associated with their host, such as fungi known to inhabit the plant phyllosphere (Porras-Alfaro & Bayman, 2011; Vorholt, 2012).
Please contemplate changing the text to: These samples represent well-preserved DNA at snapshots in time and space. For example, plant DNA bank samples not only preserve the genomic information of the targeted species, but also preserve potentially important cryptic microbial symbionts associated with the host, such as fungi known to inhabit the plant phyllosphere (Porras-Alfaro & Bayman, 2011; Vorholt, 2012).

Line 66 - 71: These increases are due in part to advances in direct environmental DNA sequencing and extrapolations based on predictions of vascular plant to fungal ratios (O’Brien et al., 2011; Taylor et al., 2014). In order to obtain more accurate estimates of true fungal diversity, increased sampling using high throughput DNA sequencing of many different types of environments is needed, and DNA banks may significantly contribute to filling this knowledge gap.
Please contemplate changing the text to: This increase is due, in part, to advances in direct environmental DNA sequencing and extrapolations based on predictions of vascular plant to fungal ratios (O’Brien et al., 2011; Taylor et al., 2014). To obtain more accurate estimates of the true fungal diversity, increased sampling using high throughput DNA sequencing of many different types of environments is needed, and DNA banks may significantly contribute to filling this knowledge gap.

Line 79: please elucidate how the 86% values was calculated (46/52*100=88%).

Line 90: please contemplate replacing was started by was settled.

Lines 92 – 95 and 98 - 101: in the reviewer´s opinion, the text is rather confusing. Please consider re-phrasing it. Also, in line 98, the authors state that all wild plants tissues harbor fungi. In the reviewer´s opinion this is also true for cultivated plants.

Lines 101 – 103: Thus, along with its banked plant samples the HPDL has also likely and coincidentally preserved a substantial portion of the diversity of Hawaiian fungi, acting as a repository for not only plant genetics, but their microbiomes as well
Please contemplate changing the text to: Thus, along with its banked plant samples, the HPDL also has likely and coincidentally preserved a substantial portion of the diversity of Hawaiian fungi, acting as a repository for not only plant genetics, but their microbiomes as well.

Lines 104: please contemplate replacing historic samples by historical samples.

Lines 120 – 132: the text written by the authors should not be included in the Material and Methods chapter. Please considering removing or moving it to other chapters (such as Introduction and/or Discussion).

Line 136: please consider removing in the DNA Library.

Lines 140 – 143, 198 – 201: the text written by the authors should not be included in the Material and Methods chapter. Please considering removing or moving it to other chapters (such as Introduction and/or Discussion).

Lines 215 – 217: in the reviewer´s opinion, the text is rather confusing. Please consider re-phrasing it.

Line 569: please include the HPDL acronym.


The manuscript includes sufficient introduction and background to demonstrate how the work fits into the broader field of knowledge. However, with the new sections included in this last version, the text lost structure. Sometimes, the reviewer had the feeling that the authors were jumping back and forward on the same subject.
Relevant prior literature was appropriately referenced.

The structure of the article conforms to an acceptable format of ‘standard sections’
Figures are relevant to the content of the article, present sufficient resolution, and appropriately described and labeled.

Experimental design

The major concerns of the reviewer were taken into account and the text was changed accordingly.

Validity of the findings

The major concerns of the reviewer were taken into account and the text was changed accordingly.

Additional comments

The reviewer acknowledges the authors ‘efforts to change the previously version of the manuscript according to the previous suggestions.

·

Basic reporting

The study shows some interesting data but the authors should refrain their comparisons between Islands, as in my opinion not enough sampling was conducted to clearly evaluate all Islands. Plant DNA banks can add some information regarding the diversity of fungi found in natural environments (qualitative data) but do not clarify questions regarding the frequency of each taxon (quantitative data).

Experimental design

It is not clarified:
1) were the plants collected using gloves and protecting the material from hand contamination? how do the authors assured that the plants had similar age, light exposure, similar humidity and environmental conditions? the same plant under different sun exposure for example may hold very distinct microbial communities!
2) how long were the leaves kept at 4˚C? How long was the DNA stored at -20˚C?
3) 1g of plant material was extracted per sample. What was the reproducibility comparing samples from the same leave? and from different leaves?
4) in my opinion UCHIME predicts chimeras much better than any other tool (Perseus or other; no questions about it). But UCHIME2 is much better than UCHIME. Please see documents and publications of Edgar et al. (the developer of the tool). The last version of the tool should be used unless there is a specific reason not to do it (better results with the oldest version for example, but anyway both need to be tested).

Validity of the findings

I do agree that the present results have some qualitative value for description of fungi on plant material. But I have serious questions regarding the quantitative analysis of the data.
"For example, we were able to confirm the distance decay of microbial community similarity from DNA bank samples collected across the Hawaiian Islands." It is my opinion that the results are not enough to reach such statements or conclusions. A single plant (or even a few plants) aren't representative of the diversity of fungi in a location or island and therefore it is not possible to compare the data at such level. the manuscript does not show enough data to support the statement that the collected samples were representative of the environment.
"We were also able to identify a temporal partitioning of phylloplane fungal communities..." the authors didn't get samples during an entire year so the authors cannot guarantee that the material they tested is representative of all temporal changes in the environment.
Parts of the manuscript based on the quantitative analysis of the data should be rewritten.

Additional comments

I agree with the authors that the present manuscript helps to clarify DNA banks as "a new and underutilized function of biological collections...". But I do not agree that such collections are enough to study the "... microbial diversity, questions regarding microbial biogeography, host specificity, and the effects of global change on microbial communities...". The plant banks can add some qualitative value but do not replace the environmental samples for quantitative analyses collected by ecologists and microbiologists over time in natural environments. In the end, any conclusion needs to be always verified in the natural environment.

---

## Round 0.3 · accepted · Accept

I have checked the new version of the manuscript and I think all comments and suggestions raised have been successfully addressed.